# Wholeness for Life and Life Eternal: A Perspective from *Ubuntu*, Paul's Reconciliation Theology, and the New Cosmology

Augustin Kassa, SMA

Department of Theology and Religious Studies, Villanova University, Villanova, PA 19085, USA; skassa01@villanova.edu

**Abstract:** The idea of cosmos unity is not recent. It has been proposed from various viewpoints throughout human history as the locus of life. To begin with, the African worldview of *Ubuntu* tells the story of life from the experience of a cosmic perspective that upholds the primacy of the community and asserts that a truly fulfilling and complete life is attainable only by those who belong to the cosmic whole. There is no 'I' without 'we'; "Because we are, I am". And, unless the "I" belongs to "we", there is no life, biological or ancestral, after death. On its part, Paul's cosmology, generally understood as proposing a two/three-tiered cosmos, has a different viewpoint when seen from a closer look. Even if he would have agreed with his religious ancestors that sin had divided reality and that diastema is to blame for the cosmos fragmentation, Paul still recounts the story of a cosmic whole. His theology of reconciliation makes that unity more evident when he suggests that through Christ, the cosmos is reconciled, and life is restored. To belong to the cosmic whole is to be reconciled with God and have eternal life. From a third perspective, today's new cosmological investigations have uncovered the unfolding story of the grand unity and complexity of the universe, which is the only locus of life known to humanity. In this universe of connectivity and entanglement, one can scientifically appreciate the absence of fragments and observe the complexified unity of all things indispensable for living. These three stories reveal that togetherness and the experience of the cosmic whole are fundamental for life and the taste of eternal life.

**Keywords:** cosmology; life; eternal life; *Ubuntu*; big bang; belonging; whole; wholeness; relationships; interdependence

## 1. Introduction

The cosmic whole is life's crucible, its vessel. It is in it, and by belonging to it, that one lives and lives eternally. This echoes "God's original intention for all creation to flourish" ([McNeil et al. 2015](), p. 22), which is re-echoed by different worldviews. Firstly, in the African worldview, life belongs to an interrupted 'circle of life'. To live, (whether biologically or spiritually as an ancestor or a spirit), to reincarnate, or to be-come, one must belong directly, interconnectedly, tangentially, and so on, to the circle of life that forms a whole with other circles of existence, making life possible. Indeed, "I am because we are, and because we are, I am". That is *Ubuntu*. Secondly, using the soteriology of reconciliation, the apostle Paul focuses on wholeness by proclaiming the gospel of life and salvation. Jesus' death reconciled the cosmos and achieved the wholeness indispensable for life. Those who belong to this cosmic whole have life and have it eternally. In a third approach, what *Ubuntu* expresses by cultural intuition and Paul proclaims as salvation in and through Jesus, science (e.g., the Big Bang cosmology and quantum physics) discovers through observation. Put differently, in regard to the emergence and development of biological life 3.8 billion years ago, its flourishment and fulfillment cannot be imagined outside the cosmic background of a 13.8-billion-year-old universe and its ultimate telos. Life and the universe stories are entangled from the beginning to their fulfillment ([Haught 2022](), p. 3).

The Nigerian novelist Chimamanda Ngozi Adichie warns us about the danger of knowing one story. In this research paper, we will know the story of life and eternal life told

from three cosmological perspectives: (1) *Ubuntu*, (2) Pauline reconciliation theology, and (3) the Big Bang cosmology. Evident in these three stories, despite their different origins, is the unambiguous perception that wholeness is indispensable for life and its abundance. While these three stories may sound alike, each of them is unique. Every cosmological application—especially those that foster division and fragmentation—is not life-giving and favorable to eternal life. While cosmology is the study of the universe as a whole, not all cosmologies are holistic, nor are their life applications. For instance, a 12th century catholic prayer (Hail Holy Queen) affirms that the Earth is the valley of tears to which humanity is banished. And suggests that it is only a pilgrim's place while our homeland is elsewhere (in heaven). Such an application of cosmological reality is not life giving, since it constitutes a danger to the Earth, a piece of the cosmos indispensable for the experience of wholeness. The same applies to religious beliefs, which retain eternal life only for their followers and promote cultures of division and the fragmentation of humanity. These are not life giving either. Despite the uniqueness of each, the *Ubuntu*, Pauline, and Big Bang cosmologies demonstrate that only holistic cosmologies, whose application in life fosters the experience of the whole and its unequivocal importance, promote life and eternal life. Therefore, each of the three stories is considered valuable and essential here; to use Christian terminology, each is salvific in its own right because of the holistic approach. As a good storyteller, no comparison is being sought. In conclusion, there is a constructive desire that all religions—specifically the Church that tends to resist the science of the new cosmology and African traditions—play a role in fending off fragmentation through integrating these life-giving cosmological stories with Paul's for the sake of the experience of the salvific life-giving whole.

Two words (κόσμος and λόγος) constitute the etymological blocks of the term cosmology. In its simplest sense, cosmology is the λόγος, the discourse on the κόσμος (cosmos). Cosmology is generally defined as the study of the universe as a whole. In a more complex approach, Raimon Panikkar distinguishes between *kosmology*—"the science . . . about *the holistic sense* of the *kosmos*" (Panikkar 2013, p. 369)—and cosmology—the scientific "objective genitive: our logos about the cosmos" (Panikkar 2013, p. 370). He characterizes the first as the scientific work of traditional societies, and the second, as the inquiries of modern science. "Neither traditional *kosmologies* nor modern science cosmologies are totally objectifiable sciences" (Panikkar 2013, p. 370). Our ancestors' and modern scientists' studies of the cosmos always converge through questions related to life, humanity, and maybe, God. For example, Paul's perception of the *kosmos* was unambiguously a question about the *anthropos* and *theos*. These three (cosmos, *anthropos*, and *theos*) are distinguishable but irreducible components that form the whole and are in themselves whole because of their relation to the whole. In the words of Panikkar, this is the *cosmotheandric* cosmology, and it "claims to possess saving powers for the wholeness (salvation) of Man, a microcosm" (Panikkar 2013, p. 370). Additionally, in the religious world of cosmologies, the cosmos, which is the home that makes life possible, is also "the body of God. Christianity likewise claims that the *kosmos* is the body of Christ and qualifies this by saying that this body is still in pangs of birth and on the way to *eschaton* that will be reached by every realized (liberated) person" (Panikkar 2013, p. 370).

Cosmological inquiries cannot be easily characterized as being similar to those in other fields. For Christopher Smeenk and George Ellis, this is because there is only one κόσμος (cosmos) that cannot be compared to anything like it (Smeenk and Ellis 2017). However, we know with certainty that this cosmos is the physical situation, the larger context that makes life, our life, possible (Smeenk and Ellis 2017). In other words, we only know, understand, and have perspectives of who we *are be-coming*, *life living* in this '*cosmosing*'. For this reason, the question about the cosmos also becomes a question about life, us, and the reasons for our religious hopes in salvation and eternal life. Consequently, in our religious aspirations, the line between the subject (who is questioning), who becomes a question, and the object questioned (the cosmos) becomes blurred because of the convergence of the cosmos, the *anthropos*, and the *theos* (Panikkar 2013, p. 370).

## 2. *Ubuntu* Cosmology

*Ubuntu* is a Bantu expression adopted by most Africans as accurately representing the worldview of the entire continent (Tutu 2011, p. 21). For John Mbiti, there is a philosophical foundation for the anthropological expression of *Ubuntu*. He writes: "[A] person does not exist all by himself: he exists because of the existence of other people. The philosophical formula about this says, 'I am because we are, and since we are therefore I am'" (Mbiti 2015, p. 108). In the meantime, anthropological and philosophical orientations would not have been possible without the somewhat accurate perception of the cosmos. The vision of the cosmos as a life-giving whole informed the African worldview and influenced the continent's civilization. From time immemorial, *Ubuntu* constituted the overarching vision from which all African institutions are built and informed (Dyer 2018, pp. 215–38, 223). *Ubuntu* has always been present in all African cultural societies and is in no way a recent creation of African scholars, as some writers tend to suggest (Dyer 2018, p. 226).[1] At the center of *Ubuntu*'s philosophical worldview is the primacy of the community, the whole upon which and without which no identity is imagined, formed, or determined (Battle 2009, Locs 28–29). Because "[c]ultures need a cosmology to understand their place in the greater framework of creation" (Frank 2011, p. 13), *Ubuntu* inspired many cosmologies in the continent.

The whole is the starting point of all parts. The Dogon people of West Africa maintain that the universe commenced from one of the tinniest *round* seeds. The *fonio* seed, known as *acha*—but scientifically classified as *Digitaria exilis*—is believed to be the smallest of all the seeds known in the region. The tiny seed containing "the potential for the existence of all reality" (Azuonye 1999, p. 48) started to experience an internal vibration that led to its expansion. For the Dogons, the expansion is achieved in seven stages, creating the archetypes of all things in the egg of the universe. The egg is then hatched, hence the birth of existence, of everything there is. A similar myth can be traced from one of the oldest civilizations on the continent. "In the Egyptian creation myths, we are told that the universe came out of a cosmic egg" (Jackson 1985, p. 9). Both the circular nature of the *fonio* seed[2] and the egg encompassing *everything* within itself suggest a compact, circular cosmos, outside of which *nothing exists*. One is born, lives, dies, and lives eternally within and never outside this circular universe originating from the cosmic egg.

The Bantu people of Congo propose a double void—*mbûngi*—to explain the relationship between the cosmos and life. In the first void, which is cosmic, "God cooked dough, the magnetic matter, the big bang" (Fu-Kiau 2001, p. 22), thus launching the universe's formation. The period is also believed to be the stage of the *kalûnga's* (the fire-force's) cooking, which is the force of the universe's expansion, the creation of new planets, and the life within them (Fu-Kiau 1994, pp. 17–34, 22–26). The Bantus had an idea of a much older cosmos because of their ancestral relationship with Ancient Egypt.[3] They maintained that the Earth is the oldest of all the planets in the cosmos. For them, the planet formation process within the expanding cosmos has three stages simultaneously corresponding to three different *circular* layers within it (the universe). The outer red layer is made of red planets. These are still in the process of formation, fusing with others as they crystallize and cool down. Life as we know it is nonexistent on those planets. Gray planets in the gray middle *circular* zone, on the other hand, are fully formed. But then,

> These planets are naked, dry, and covered with dust. Gray planets are without life as we know it; i.e., they are without plants, animals, and, of course, without human beings. The Bântu-Kôngo teaching suggests that if left alone, these planets will eventually complete the four stages of the planet transformation process, (. . .and) see the rise of plants, animals, and beings like humans share life on them (Fu-Kiau 2001, p. 24)

Within the third and inner circle, one finds green planets, the breathing planets (Fu-Kiau 2001, p. 24). Bantus think there may be many other green planets like the Earth, which are unknown to us. These would have already passed through the first two stages

and reached the third stage of planet formation. However, the fourth stage, that of life as we know it, is, for now, something peculiar to planet Earth. The Bantus are ready to acknowledge that other life forms that differ from what we know may exist. What that means remains elusive. But within the mini void of planet Earth, the fire-force (*kalûnga*)[4], complete by itself, emerges (Fu-Kiau 2001, p. 19). If it was cooked initially, it now appears as a whole reality. Firing up the vacuum, it overcomes it, and life as known to us emerges. "Because *kalûnga* was the complete life, everything in touch with the Earth shared that life after itself. That life appeared on the Earth under all kinds of sizes and forms: Plants, insects, animals, rocks, human beings, etc." (Fu-Kiau 2001, p. 21). Bântu-Kôngos also think that if life is to appear on the gray planets—like Mars and the moon—it will be facilitated by the firing up of the *kalûnga*. The *kalûnga* is part of the universe at its beginning and is present in and with it throughout its expansion and its becoming, just as it is responsible for the emergence of life.

Life, fire-force, life-force (*kalûnga*) is, at the same time, one and distinct, depending on its manifestation. Father Placide Tempels, a Belgian Franciscan missionary, explains in *Bantu philosophy* that the *kalûnga* is Africa's supreme value. "This suprem [*sic*] value is *life, force, to live strongly,* or *vital force*" (Tempels 2010, p. 44). The belief is that there is only one life or vital force that connects and keeps the whole cosmos connected. To clarify, Bénézet Bujo writes:

> God is the dispenser of life. But...in Black Africa's concept ... Life is a participation in God, but it is always mediated by one standing above the recipient ...This hierarchy belongs both to the invisible and to the visible world. In the invisible world, the highest place is occupied by God, the source of life. Then comes the founding fathers of clans, who participate most fully in the life of God. Then comes the tribal heroes, deceased elders, other dead members of the family, and various invisible beings, including earthly powers, although these belong partly also to the visible world...Then comes beings belonging to the visible world. They include the king and the queen-mother, as well as those who wield or represent the royal power; chiefs of the clan and the oldest members of families; heads of households; and family members (Bujo 2006, p. 20)

Bujo understands life or life-force as following a specific pattern. It is an internal cosmic law of togetherness as it flows from God, mediated through the dead and living ancestors, and passed through the community and the family to the recipient. The life force bridges space and matter, but also time, which will be discussed below. Father Tempels records a nuanced similarity. His experience of the same people reflects a vision of various life's manifestations at different stages, thus covering a more extensive spectrum. He writes: "[A]ll beings in the universe *possess vital force of their own*: human, animal, vegetable, or inanimate". However, "[e]ach being has been endowed by God with a certain force, capable of strengthening the vital energy of the strongest being of all creation: man" (Tempels 2010, p. 46). This should not be interpreted as anthropocentrism, because Africans believe in the sacredness of all creation (Maathai 2007, pp. 5–6). Instead, the objective is the responsibility, care, and sustainability necessary for maintaining the one continuum of life and harmony in the cosmos. In a nutshell, life, or life-force, is only possible within the universe, in which everything is connected. The one who wishes to have life can only ensure that s/he belongs to this cosmic community where life is transferred, communicated, shared, and strengthened. This is *Ubuntu*.

No one can effectively discuss cosmology without taking seriously the relationship between time, space, and matter. The above paragraphs have explored the relationship between space, matter, and time as well, since these are difficult to separate in African cosmology. Nevertheless, we may now turn our attention to discussing the notion of time, which is circular in African cosmology, and its implication on matter and space. Time runs circularly at two interwoven levels. The first is cosmic, and the second is event-related time.

> [T]ime is both abstract and concrete. At the abstract level, time has no beginning or end. It exists on its own and flows by itself, on its own accord. Yet, at the concrete level, it is *dunga* (events) that make time perceptible, providing the unending flow of time... (Fu-Kiau 1994, p. 20)

The cyclic nature of time in African cosmology does not indicate that it only repeats itself. Instead, because cosmic time is deeply entangled with the events-related time at the Big Bang, "the beginning of all time" (Fu-Kiau 1994, p. 22), the African notion of time is primarily pedagogical.

> [T]ime lies at the core of our understanding of not only the universe and its processes (dingo-dingo) of creation, transformation, and functioning but also of life itself and its functioning. It is through time that both nature and man become comprehensible to us (Fu-Kiau 1994, p. 20)

Time is a tool that teaches about the importance of the whole, whose origin is related to the cosmic time and the events at its beginning. But so are also the associated events of the cosmos' expanding processes, that is, its continuous formation, which includes cooling, crystallization, and the apparition of microscopic beings tied with the sun's rising, resulting in humanity's emergence as the process continues.

The green planet's maturity stage is the most essential phase of time's pedagogical importance. This is considered the time of collective growth and maturation (Fu-Kiau 2001, p. 27). Before humanity's emergence, the building blocks of humanity's genetic code, to use a scientific language, was/is one of wholeness. In the Bantu cosmology, it is expressed as "the 'V of life'" (Fu-Kiau 2001, p. 28), which is a growth concept. It stipulates that standing tall through life's circle(s), the *anthropos* must recognize in and through the self the absolute oneness between "the earth and the sky, the upper and lower world" (Fu-Kiau 2001, p. 28). In the 'V of life', the self that crosses the circles of existence from birth to eternity and back to birth for those who reincarnate must find and give meaning to the cosmos that gives sense and life to who the self is and without whom it is not. Thus, humanity is genetically coded to be informed before it comes into existence that its *be-coming* and the *be-coming* of the cosmos are entangled, and they together form one whole. Therefore, it is not surprising that "*Ubuntu* implies a constant awareness that an individual's actions today reflect the past and will have far-reaching consequences for the future. A person with *Ubuntu* knows his or her place in the universe..." (Broodryk 2010, p. 43). This cosmic code (*Ubuntu*), built into the genetics of the *anthropos*, prepares humanity when it emerges to the truth of its interlocking reliance and intimate connection to past and present existence. The *anthropos* must also be aware that the cosmic future, which includes them physically or otherwise, relies upon them. In a nutshell, when s/he arises, the *anthropos* must know the significance and necessity to maintain and sustain the interconnectedness of the whole if s/he is to *be*-come, live, and flourish. As Panikkar writes, "Whatever the temporal origin of Man may be, the biological genesis of a thing does not disclose the essence of the thing. *How* an entity has come to be following a *linear temporal* sequence does not disclose what the entity is" (Panikkar 2013, p. 293). We come from very far, from within a complex system of entangled circular relationships. We form a whole with this system of intertwined circular relationships and are headed very far in the same, maybe to a more complexified reality of *Ubuntu*, of togetherness.

The cosmos has always known how to live *Ubuntu*; that is what it has always done, in and through the undeniable interlocking relationship between space, time, and matter. The human being, however, even though encoded with *Ubuntu*'s genes is, so to speak, new in its (*Ubuntu*) process because of its (humanity's) relatively recent appearance in the history of the cosmos and must be initiated to respond to his/her genetic nature. Consequently, Africans are trained and educated to know that a person must be with *Ubuntu*; that is, s/he must be open and available, s/he must not feel threatened, and must have a proper self-assurance that comes from knowing that s/he belongs in a greater whole (Battle 2009,

p. 2). The South African Anglican Bishop Desmond Tutu clarifies *Ubuntu*'s educational and anthropologic reality for the human family and writes:

> "A person is a person through other persons". We need other human beings for us to learn how to be human, for none of us comes fully formed into the world. We would not know how to talk, to walk, to think, to eat as human beings unless we learned how to do these things from other human beings. For us, the solitary human being is a contradiction in terms. *Ubuntu* is the essence of being human. It speaks of how my humanity is caught up and bound up inextricably with yours. It says, . . . "I am because I belong". I need other human beings in order to be human. The completely self-sufficient human being is subhuman (Tutu 2011, pp. 21–22)

It is worth noting that when Africa speaks of a human being, it is not limited to the living. Deceased forebears and the living spirits, as outlined by Bujo earlier, are integral components of the equation. Johann Broodryk confirms it when he writes:

""Persons" includes not only living human beings, but ancestors who have already died and children who have not yet been born. *Ubuntu* embodies a deep respect for ancestors, and includes all the attitudes and behaviors necessary not only for a harmonious life with other individuals on Earth but with ancestors in the world beyond death and with those who will live on Earth in the future. Every individual is the fruit of his or her ancestors and will become the ancestor of all future descendants" (Broodryk 2010, p. 43).

Of significance here is the continuous interplay between time, space, and matter as it unfolds with precision, maintaining a substantial connection between the biological existence, encompassing "life and its creative energy (reproduction)" (Fu-Kiau 1994, p. 26), and the cosmic living presence of the "deceased" ancestors and spirits. The end of the biological cycle works like a dam that propels one back into the cosmic circle of life with the possible capacity to bring one back into the essential circle for a fresh beginning. Thus, things

> perish in order to change and begin a new cycle. . . Dying is not only a process but also a 'dam of time'. As a dam of time, it has its own landmark on the timeline path, and as a process, it permits life to flow and regenerate . . . its power/energy . . . to create a new state of being or undergo a transformation capable of rejoining the body of the universal 'body-energy'. The living energy that existed before becoming living matter at conception is then freed again (Fu-Kiau 1994, p. 27)

In the African worldview, nothing is separate; everything—time to space, space to matter—is connected to form a whole in a vast circle of other interlocking, interacting, overlapping, and intersecting circles. Life belongs here in this cosmos, outside of which we have no life, because together with it, we form one whole reality. To live physically or spiritually is to belong, to be connected to the whole we are part of, which makes us who we are. Tutu writes:

> *I am because we are*, for we are made for togetherness, for family. *We are made for complementarity. We are created for a delicate network of relationships, of interdependence with our fellow human beings, with the rest of creation* (Tutu 2011, p. 22)

Simply stated, without God, ancestors, the environment, and the cosmos, I am not and will not be without the whole. Similarly, the whole is not the whole without me or any of its parts. To *be-come* is to be connected, to remain in, to belong to the whole. In the African worldview, there is no moment when it is acceptable, permissible, or even possible to *be-come* outside the whole. In *Ubuntu*, belonging and togetherness in the whole and the formation of the whole govern life and form the lenses for the reason of our hope (salvation). Jesus is only possible because he is situated in this existing and indispensable circle of life. Thus, he emerges in Africa from an already *Christ*-soaked, reconciled, cosmic

whole (Cox 2021, pp. 1–6, 3; Rohr 2019, p. 15) to point us to what we have always known. Life belongs to the whole, which is its crucible.

### 3. Paul's Theology of Reconciliation

Life indeed belongs to the whole. The apostle Paul centralizes the Christ event as the catalyst for cosmic wholeness throughout his writings. Paul's insights stem from the cosmological thoughts of the Jewish and Greco–Roman worlds he lived in. Israel's cosmology, which emerged from religious practices (Adams 2008, pp. 5–27, 20), understood the tabernacle, or temple, as God's vision of the cosmos. The Hebrew Bible writers were not theological idealists (Davis 2008, p. 3); they were pragmatic writers reflecting on the relationships between human beings and the material sources of life as essential elements to living and being in the presence of God (Davis 2008, p. 3). For them, God, in the beginning, established an order (*Shalom*) indispensable for life. However, sin in the created cosmos, which is not inherently evil (Adams 2008, p. 24), has fragmented God's masterpiece. And now, the Earth suffers due to people's sins and injustice.[5]

The wedge introduced by sin must be removed for the universe to be whole again. A new cosmos is reimagined that could foster into "our minds a fresh vision of the world as God's creation" (Davis 2008, p. 143). The primordial universe, that is, Eden, its garden, and all within it, was God's original sanctuary or 'temple' (Pitre and Bergsma 2018, p. 102). The new cosmos must equally reflect this reality. The instructions in Ex 25–31 and the building in Ex 35–40 of the tabernacle became the blueprints of the new cosmic vision. Ellen Davis opines: "The wildness sanctuary is a microcosm, an image of the world viewed from Sinai" (Davis 2008, p. 143). Edward Adams observes the same, and remarks that the construction of the tabernacle is recognizably "suggesting 'a homology of world building and temple building'" (Adams 2008, p. 20). The temple is built to mimic the original cosmic wholeness, with God, humanity, and the Earth (trees, gold, onyx, bdellium, etc.) present. The harmony encapsulated in the well-ordered labor and relationships reimprints the vital *shalom* essential for life to flourish. And from this new cosmic whole flows the river of life that gives life (Ezekiel 47).

The enchanted Greco–Roman world's cosmology also influenced Paul. The heavenly forces, the personified forces of good and evil, and angels and demons enchanted the cosmos, making it whole and dynamic (Norman 1992, p. 126). Hence, the cosmos, besides being designated 'heaven', *οὐρανος,* is identified as "'the whole', *το ολον,* 'the all', *το παν,* and 'all things', *πάντα*" (Adams 2008, pp. 6–7). Adams' remarqued that the "Stoic worldview was the most influential in Greco-Roman antiquity" (Adams 2008, p. 16; Hahm 1977, p. xiii), and suggested that these Stoics "viewed the cosmos biologically . . . It comprises body and soul and is animated by 'breath' (*πνευμα*). 'Breath' is the life-force of the cosmos, sustaining it and maintaining its unity. The cosmos has birth and growth, but it 'must be not said to die'..." (Adams 2008, p. 17). Alive and eternal, the cosmos is still perceived as created but coextensive with the divine.

> Stoics viewed the cosmos as the well-constructed product of a divine creator. They differed from Plato and Aristotle, though, in making the divine intelligence, 'god', coextensive with the cosmos. 'God' was understood as the rational, active principle—the *logos*—present in matter (and inseparable from it), pervading it and giving it order (Diogenes Laertius 7.134) (Adams 2008, p. 16).

God's presence in matter paved the way for human beings to be an extension of the gods (Norman 1992, p. 127). Thus, the cosmic whole is formed by the cosmos *and the anthropos,* coextensions of the *theos.* This *theo–cosmo–anthropo* whole is where the life force, 'breath' (*πνευμα*), and life reside.

Besides the Jewish and Greco–Roman cosmological intellectual background of Paul, the Christ's event with the cross at its center for him is equally a cosmological occurrence. The event removes the wedge of sin and unites the whole. Unlike those who see the cross' event as another dividing—stumbling or destabilizing—block (1 Cor 1: 23), Paul understands it as a bridge that brings the whole back together and even makes the resurrection

possible. Christ, in whom all the fullness was pleased to dwell, connects Jews and Gentiles (1 Cor 1:24) and reconciles *all things* (Col 1:19–20). Paul, therefore, sees the cross as the fiber, the mortar, the capstone binding humanity, and, with humanity, 'all things' in the cosmos together for life and life eternal, exemplified in the resurrection of Christ. Christ's event is the event of newness that brings even opposites to belong together for the sake of the whole, the vessel of life. Here, the stumbling block and capstone work hand in hand, weakness and power embrace, wisdom and foolishness kiss, and death and life encounter as God and humanity meet (1 Cor 1:18–29). In Jesus, the man–God, dead–alive symbols of reconciliation and harmony, the promise of humanity's flourishing as the *homo-deus* is expressed as a symbiotic relationship between man–God and death–life.

The use of the names (1) heaven, (2) earth, and (3) midair, etc., in his writings, reveal Paul's awareness that the consciousness of division remains despite the reconciliation achieved by Christ. While humanity gradually progresses towards the unbroken conscious experience of cosmic wholeness, the nomenclature is in order (Panikkar 2013, p. 26). Paul is not reinventing the wheel. Plato, in the Timaeus, characterized time as "a moving image of eternity" (Seissl 2022, pp. 1–28), which later, in the Stoic tradition, was translated to (διάστημα) diastema (White 1996, pp. 183–98, 189; Durand et al. 2023), meaning spacing. The concepts allow for creation and humanity to be connected to the origin and distanced from it in time and space. As noted by Urs Von Balthasar, Gregory of Nyssa remarked that one can have a glimpse of humanity's origin. Still, diastasis, a human condition absent in God, hides that origin from the *anthropos* (Balthasar 1995, p. 28).[6] Diastema, for Stoics, was the result of being a creature. Paul indeed agreed by emphasizing God's transcendence and *moreness*, neatly held together with the cosmic whole that humanity cannot fully capture (Rom 11:33–36). Paul, however, believes that despite diastasis, humanity can have a glimpse of the eternal whole. This affords him the methodological gymnastics one sees in his letters. In these, moving within the life-giving cosmic whole, he navigates back and forth between origin and eternal future hope through time and space. In the meantime, Jesus, who came at the fulness of time (πλήρωμα τοῦ χρόνου) (Gal 4:4) and reconciled *all things*, now helps to master diastema until we come to the full experience of the cosmic whole.

For Paul, God, through the divine *Logos* Christ (πρωτότοκος),[7] who is the firstborn, created *all things*. And the created cosmos (τὰ πάντα) belongs to God because it is from him, through him, and in him (see in Rom 11:36, 1 Cor 8:4; Col 1:16). However, the location of Paul's cosmos in God does not limit its dynamism, since he maintains that God can and still calls what does not exist into being. "God . . . gives life to the dead and calls into existence the things that do not exist" (Rom 4:17). The cosmos, which is from God, through God, and in God forming a whole with him, is life giving, even for what is or seems dead. Indeed, there cannot be death in the cosmic whole in and with God. This elucidates why the goal of the 'new things' called into existence in the cosmic whole cannot and must not be separate from that of the whole, which is life, since it belongs to the whole. The already existing old and the new get bound together in God to form the life-giving whole. The unity between God and the cosmos, even as the universe expands and God with it since it is in God, promises and always gives life to those who belong.

## 4. New Cosmology

As we shift our focus from the central role of the community in African cosmology and proceed through Paul's reconciliation soteriology, which served to describe cosmic wholeness as the crucible for both biological and eternal life and belonging as achieving both salvation and life everlasting, we now turn our attention to the findings of modern science and technology. The Big Bang cosmology's emphasis on the origin of the universe and the inherent presence of energy in reality, coupled with quantum physics' exploration of energy's diverse forms and its interconnection with our consciousness illuminates the idea that life and human flourishing can only manifest within the cosmos. John Haught notes that the natural world "after giving rise to life and mind. . . is just emerging from the dark womb of its past to an unpredictable future" (Haught 2022, p. 7). According to him,

this should matter to science, theology, and every culture. For many centuries, humanity has speculated on the nature of the universe as a compass for understanding reality and its implications for us. Time and cultures shape cosmologies, just as cosmologies are shaped by cultures (Frank 2011, p. 17). But life emerged from a sea of energy and is moving towards a future in which womb vision remains obscure and unpredictable, because the cosmos is still awakening.

However, time, cosmological speculations, and scientific developments sometimes converge to generate a once-in-a-generation, life-altering novelty that allows us to see what was in the womb of the cosmic past that gave birth to the presently known cosmos. The knowledge then opens the path to inferring, knowing, never definitively, but within a broad spectrum of probable possibilities, what may be in the womb of the cosmic future from the echography of the present womb.

The paradigm shift in humanity's understanding of the cosmos commenced in 1543, when Nicholas Copernicus discovered that the Earth was not the center of the solar system. This time, the investigation to probe what is really going on with the universe will not be guided by myths, but by scientific observations—sometimes speculative—aided by technological advances. And so, in 1918, the American astronomer Harlow Shapley noticed that not even our galaxy had the sun at its center. A little later, between 1923 and 1929, Edwin Hubble discovered that our galaxy, the Milky Way, was just one among countless other galaxies of the universe. And for the first time, humanity was scientifically coming to a solemn awakening. The *anthropos* and our planet were not at the center of anything. The universe was still expanding, and galaxies were moving away from one another, with those far from ours moving faster than the closer ones (Delio 2020, p. 2).

Almost in the same period, in 1931, George Lemaître, a Belgian Catholic priest, made groundbreaking observations. In a paper entitled "A Homogeneous Universe of Constant Mass and Increasing Radius Accounting for the Radial Velocity of Extra-galactic Nebulæ", Lemaître suggested that the cosmos came from "the Cosmic Egg" (Lemaître 1931, pp. 489–90). He expanded the idea at the British Association in London in the same year when discussing the relationship between the physical universe and spirituality. He articulated the possibility of an initial point or a primeval atom, "the Cosmic Egg, exploding at the moment of the creation".[8] Following quantum physics, which we shall explore below, a theoretical explanation of Lemaître's proposition suggests that the cosmos commenced in a quantum vacuum. In and through the expansion process "set up by radiation itself . . . at the starting point" (Lemaître 193, p. 489) there was simultaneously an internal process of granulation and criticization, leading to the formation of atoms and stars. These realities developed through the emergence of hydrogen and helium atoms facilitated by the presence of nuclei. Whereas Lemaître concludes that "the largest part of the universe is forever out of our reach" (Lemaître 193, p. 489), at least we know that the impact of the collisions of stars and asteroids led to the creation of the planets, the Earth we inhabit included. On our planet, the formation of the foundational elements (carbon, oxygen, nitrogen, and phosphorus), essential for the organic form of life recognizable to us today, was made possible by energy and fusion processes.

In the second half of the 20th century, two scientists, Arno Allen Penzias and Robert Woodrow Wilson, finally discovered the evidence of a 13.8-billion-year-old universe. Professor Ilia Delio explains that "in 1964. . .two scientists working at the Bell laboratory in New Jersey discovered 'cosmic microwave background' that was left over from the beginning of the universe more than 13 billion years ago" (Delio 2020, p. 3). However, Lawrence M. Krauss suggests that the accuracy of the measurement was only achieved in 2006, thanks to the WMAP satellite that helped observers to accurately "measure the time since the Big Bang" (Krauss 2013, p. 124). Put together, all these discoveries affirm a vast cosmos in which everything is connected, at least from the perspective of the starting point and its expanding energy.

Associated with these cosmological discoveries is the birth of quantum physics. Max Karl Ernst Planck took as the object of his research the understanding of the "ultraviolet

catastrophe": the correlation between energy, frequency, and light color. In a scientific world in which it was widely accepted that light behaved in a wavy manner, the photoelectric effect and the ultraviolet catastrophe proved challenging to explain. In 1900, however, Max Planck proposed that "electromagnetic energy could be emitted only in quantized form, in other words, the energy could only be a multiple of an elementary unit: *E = hν*" (Max Planck 2023; Planck's Constant 2023). With this formula, Planck opened the unchartered, mind-bending world of quantum physics. In 1905, Albert Einstein introduced the concept of quanta to elucidate the photoelectric effect, proposing that light was composed of minute particles. During this period, the traditional perception of light as exhibiting a wavelike behavior began to coexist with the emerging recognition of its particle-like nature (the wave–particle nature of reality). The experiments and conclusions drawn from light behaving simultaneously with a particle and wave were also observed in subatomic particle movements. The basic understanding of nature's reality was beginning to be challenged. As dark matter and energy became part of the scientific observations explaining the stars' orbital movements and the universe's acceleration (Currivan and Laszlo 2017, p. 6), it became increasingly apparent that all is interconnected. The cosmos is one and whole, with all reality within it inherently held and connected by energy matter and its properties.

A question that arises is, if the universe is connected, forming a whole, then why is our perception of it different? Science postulates neither sin nor diastasis. Instead, Niels Bohr and Einstein engaged in one of the most consequential conversations of our era regarding the understanding of the connectedness between the cosmos, matter, and consciousness. Einstein explained that observation does not fix reality, but instead provides information to our knowledge. He is anecdotally remembered to have asked Abraham Pais, his biographer, if he really believes that the moon exists only when we look at it. He also rejected the principle of a hidden variable through which one particle's observation and/or behavior can affect another (Bohm 2002, p. 93). Similarly, he dismissed the principle of nonlocality as a 'spooky action at a distance' (Currivan and Laszlo 2017, p.7). For him, such a distant, interactive relationship would require information to travel faster than the speed of light, which is unacceptable.

On the other hand, Niels Bohr saw reality differently. Holding onto the principle of entanglement suggests that reality is but the fruit of our observation. Every reality is a wave and consequently immense, and as such, possesses an infinite spectrum of probable possibilities. Observation is what fixes reality to this or that aspect. Biological life and death is, in that sense, a matter of observation. Erwin Schrödinger's experiment of a cat in an opaque box with radioactive material underlies the essential role of observation. The radioactive material, having a 50% chance of being released provides a fifty–fifty chance for the cat to be dead or alive. According to Schrödinger, direct observation is the only way to know if the cat is alive or dead. Since the cage is opaque, and no one can see inside unless opened for observation, mainstream quantum theory concludes that "the cat is *both* alive and dead. He [the cat] exists in a superimposed state of both conditions at once. . ." (Zohar 1991, p. 39). In humanity and everyday vocabulary, curiosity is what kills the cat. Otherwise, we are always alive. And the capacity to always observe life and live is what our Christian religion calls salvation. But this life is only possible if we consider the whole and belong to it instead of the part that can give the impression of an end (death).

Bohr was in tune with the interconnected nature of things at the subatomic level of existence. According to David Bohm, Bohr "argued that in the quantum domain, the procedure by which we analyze classical systems into interacting parts breaks down, for whenever two entities combine to form a single system (even if only for a limited period of time) the process by which they do this is not divisible" (Bohm 2002, p. 93). Life and the cosmos cannot be divided. Because once in history, they interacted when the cosmic wholeness favored the emergence of life. Additionally, there is an unbreakable unity of energy at the subatomic level, constituting the base of all things. Life and the cosmos are headed together, with life going where the cosmos is going. Thus, for Bohr, observation does not tell us as much about the fragmented system we wish to observe as it does about

*itself as a whole or part of a whole* (Bohm 2002, p. 94). Reality is indivisible. "It is evident that according to Bohr's interpretation, nothing is measured in the quantum domain . . . Hence, there is no meaning to the supposition that there was something there to be disturbed in the first place" (Bohm 2002, p. 96).

Bohm agrees with Bohr and characterizes reality as "unbroken wholeness of the totality of existence as an undivided flowing movement without borders" (Bohm 2002, p. 242). For him, "what we call empty space contains an immense background of energy, and matter as we know it is a small, 'quantized wavelike excitation on top of this background, rather like a tiny ripple on a vast sea" (Bohm 2002, p. 242). Bohm thus introduced the idea of the three-dimensional perception of the universe: the holographic universe. Bohm explains that Greek philosophers Zeno and Parmenides' idea of a *plenum* universe should not be understood in Newtonian terms as the 'void' being filled with material particles like atoms or ether (Bohm 2002, p. 242).

> Rather, one is to begin with the holomovement, in which there is an immense 'sea' of energy. . . understood in terms of a multidimensional implicate order . . .while the entire universe of matter. . .is to be treated as a comparatively small pattern of excitation (Bohm 2002, p. 243)

The implications of Bohm's finding are profound. When we trace the trajectory of the cosmos from the Big Bang to the present moment, it becomes clear that the 13.8-billion-year-old universe, along with everything contained within it, is part of this vast 'ocean' of energy, marked by occasional tiny excitation patterns.

> No subsystem, whether an elementary particle, a person, a planet, or a galactic cluster within our universe, is or can be completely isolated. Everything at all scales of existence is being progressively discovered to be inherently related by in-formational content, flows and processes. . . (Currivan and Laszlo 2017, p. 35)

Because of the inherent flow of energy, information, and processes, our planet and all within it belong to the whole. And we cannot and will never be able to claim the existence of life, whether now or in the 'future', physical and/or spiritual, outside of this energy-filled cosmic whole, which constitutes the baseline of every excitation. In the long run, if *Ubuntu* and Paul say we must belong to the cosmic whole to be saved and live eternally, quantum physics proves that cosmic wholeness is the vessel that contains life. But because we belonged to the cosmic wholeness at some point, thanks to our interaction with it at that point and the same energy background we share, we will always belong, be saved, and live eternally.

## 5. Conclusions

In Africa, when one says, "I am because we are and because we are, I am", (*Ubuntu*) s/he expresses awareness about the centrality of the community of all communities (human, environmental, spiritual, divine, etc.) necessary for the sprouting, growth, sharing and survival of life. The 'being' of each of these 'communities' of the cosmos and their interdependent relationships are vital for each system's well-being and life and is the basis for the life of the *anthropos*. Paul's theological perspective reconstructs the whole through the reconciliation of the cosmos achieved in and through Christ. Within God is the cosmos, and with God forming the whole, it breathes the life-force offered to all who belong. "We beg, on Christ's behalf, be reconciled to God". Everyone is needed for the whole to be whole. So, Paul invites the communities and their members to love, obey, care for the other, etc., thus inviting all to belong to the visible micro-whole: the Church, the body of Christ. Objectively speaking, despite being whole because of her relationship to the whole, the Church is just part of the whole and would be dead unless she, too, makes a conscious effort to belong to the macrocosmic whole. Through attention to the universe's beginning, its energy, the observation of subatomic particles, their interactions, and their relationship to our minds, the Big Bang cosmology and quantum physics show that the undivided whole of the cosmos is the locus of life and living. In telling these three life stories, we

realize that life and its fullness are based on the search and experience of the whole. Thus, we are all invited, particularly the Church entrusted with the ministry of reconciliation and of change, to learn to constantly morph reconciling fragments to form the greater whole necessary for eternal life.

**Funding:** This research received no external funding.

**Institutional Review Board Statement:** Not applicable.

**Informed Consent Statement:** Not applicable.

**Data Availability Statement:** Data are not contained within this article. Data sharing if not applicatble.

**Conflicts of Interest:** The author declares no conflict of interest.

## Notes

[1] "The oneness of Egyptian and Black culture could not be stated more clearly. Because of this essential Identity of genius, culture, and race, today all negroes can legitimately trace their culture to ancient Egypt and build a modern culture on that foundation" (Diop 1989, p. 140).

[2] Traditional Indian cultures also understand the cosmos to inhabit the seed. "In the seed is the cosmos". see Vandana Shiva, TikTok video: *About the seeds and Nine Planets.* https://www.youtube.com/shorts/POzOBxz2x5w (accessed on 22 August 2023).

[3] There is a map that describes the double contact that exists between Egypt and the continent (Diop 1988, p. 218).

[4] *Kalûnga* has multiple meanings in the Bantu cosmology. It also embodies the idea of immensity—that which cannot be measured—source and the origin of life, the principle god-of-change, the principle that continuously generates, life force, and the balancing plane line of all energies.

[5] Genesis 4: 10 describes the earth crying out to God to ask for justice. Passages like Amos 1:2, Jeremiah 23:9, etc., describe the earth mourning (Moo and Moo 2018, p. 106).

[6] We need to note that Plato's definition was much more targeted on time being the number that gives an account for the change of the cosmos. But Gregory uses it to maintain a link, a relationship between the immutability of God and humanity's mutability.

[7] Rabbis Bechai called YHWH Himself "Firstborn of the World". Πρωτότοκος (translated: 'the firstborn') does, consequently, not render Christ less than God as Arius argued when he used the expression. Here, Paul deploys it to describe both priority in time and supremacy in rank, in which case, Paul's consideration of the beginning of time is both prelapsarian and postlapsarian, but with moments of great excitations. The Christ event in its totality is one of those moments.

[8] https://www.biblegematria.com/the-big-bang.html (accessed on 15 August 2023).

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
