# Peer review of "Wholeness for Life and Life Eternal: A Perspective from Ubuntu, Paul’s Reconciliation Theology, and the New Cosmology"

_religions, doi:10.3390/rel15020202_

Round 1

Reviewer 1 Report

Comments and Suggestions for Authors

In academic writing, it is always important to establish a clear methodological approach to the study. In this paper, that is missing entirely – the author needs to explain his research approach and methods.  

Very importantly, the author should offer a brief outline of the paper towards his/her conclusion. That gives the reader an insight into the major themes that would be discussed leading to the conclusions.

One outstanding difficulty of the work is that the meaning of Ubuntu, especially as it is relevant to the discussion seems to be presumed: To exemplify this see this paragraph

“This worldview is as old as the continent and its civilization and constitutes the overarching vision from which all African institutions are built and informed since time immemorial.12 Consequently, Ubuntu has always been present in all African cultural societies and is in no way a recent creation of African scholars, as some writers tend to suggest.13 At the center of this worldview is the primacy of the community, the whole upon which and without which no identity is imagined, formed, or determined.14 Because “cultures need a cosmology to understand their place in the greater framework of creation,”15 Ubuntu inspired many cosmologies in the continent.”

The paragraph should rather start by explaining Ubuntu as a philosophical anthropology and show how the philosophy encompasses/applies to varied aspects of life as it relates to being. This done, one can now give examples from various Africa societies to show that it is a philosophy that is present across the continent.

Similarly, Cosmology has not been explained in its rights in any way in the entire work.

An explanation of the major conceptions and a discourse on how they interconnect - on what ground is it academically possible to study these concepts together - A brief explanation of the meeting point of these three major conceptions discussed here would enrich the text. 

The author needs to reference African writers, especially pioneers of African theology who introduced Ubuntu into academic scholarship like John Samuel Mbiti, and other later works like Murovo, “Ubuntu,” Diogenes 59, no.3-4 (2014).

Revision of sentence constructions, especially to improve the flow of argument from one paragraph to another is also needed.  

Comments on the Quality of English Language

The author needs to revise most sentences to make meaning as they relate to preceding or succeeding sentences/paragraphs. E.g, In the introduction, the author talks about

“In a third approach, what Ubuntu expresses by cultural intuition and Paul proclaims as salvation in and through Jesus, science …” There was nowhere in the preceding sentences that talked about 1st and 2nd approaches. Readers cannot presume …

When starting a new paragraph, one cannot ‘exemplify’ without introducing the subject of discussion it is seen in this paragraph:

“To exemplify, the Dogons people of West Africa maintain that the universe com-menced from one of the tinniest round seeds. The fonio seed, known as acha – but scientif-ically classified as Digitaria exilis – is believed to be the smallest of all the seeds known in the region. The tiny seed containing “the potential for the existence of all reality”16 started to experience an internal vibration that led to its expansion. For the Dogons, the expansion is done in seven stages, creating the archetypes of all things in the egg of the universe. The egg is then hatched, hence, the birth of existence; of everything there is. A similar myth can be traced from one of the oldest civilizations on the continent. “In the Egyptian crea-tion myths, we are told that the universe came out of a cosmic egg.”17 Both the circular nature of the fonio seed18 and the egg encompassing everything within itself suggest a com-pact, circular cosmos outside which nothing exists. One is born, lives, dies, and lives eter-nally within and never outside this circular universe originating from the cosmic egg.”

Author Response

Thank you for the constructive observations and comments. I am currently addressing this. I will (a) include a short paragraph on the methodology, (b) a sentence to define cosmology, (c) introduce John Mbiti to the conversation, and (d) address the English.

  1. However, I mention various times that I am telling three stories of life and salvation. One can notice a similarity between these three stories: that of wholeness. As the Nigerian novelist Chimamanda Ngozi Adichie maintains, there is a danger in only knowing one story. But there is a virtue in knowing that some stories, even though from different horizons, do sound alike. I am simply drawing attention to the fact that these stories have a common denominator: wholeness is necessary for life and salvation. The observation is absolutely different from the general perception that salvation is achieved through separation fragmentations, as exemplified in “extra ecclesiam nulla salus.”
  2. I am indeed writing a paper on life and eternal life, focusing on wholeness. I am not mainly writing a paper on Ubuntu or an introduction to Ubuntu. So, I am presuming that the reader has an idea, at least a general one, about Ubuntu when approaching the paper.

Reviewer 2 Report

Comments and Suggestions for Authors

General comments:

This is a really interesting paper. The exposition of the African worldview of Ubuntu is fascinating, and the link with the theology of Paul is novel. I’m less certain about the section on scientific worldviews. It clearly isn’t possible to say much about cosmology from a scientific point of view in the space of less than a third of an academic paper, but this makes it difficult to offer more than a few generalisations from a wide-ranging field. I’m inclined to think that it might be better to focus on the first two sections, perhaps saying a bit more about the link between them, and to reserve the material on cosmology from a scientific point of view for another project.

Specific comments:

Page 2, paragraph 1: ‘Science’ is followed by ‘logos’ in brackets, but ‘logos’ means ‘word’ and it’s not clear what the link between that and science is. It might be better to put ‘logos’ after ‘discourse’.

Page 8, paragraph 2: ‘Edward Adams’ appears in full for the second time. The usual convention is to give the full name as it appears on the publication at the first citation and then just the last name at each subsequent citation.

Page 8, paragraph 2, ‘according to whom D Hahn had suggested that’: This is rather a long sentence which includes both a secondary citation and an argument. I think it would be preferable to cite the publication yourself if you can. If you can’t obtain the source, or it’s important for you to note that Adams has highlighted this, it would probably be better to put this in a sentence on its own.

Page 9, paragraph 1: ‘fibber’ should presumably be ‘fibre’.

Page 9, last paragraph: ‘live-giving’ should be ‘life-giving’.

Page 12, second paragraph, ‘Einstein proclaims loud and clear to whoever would listen’: This doesn’t sound like evidence/an argument.  

Page 12, last paragraph, ‘curiosity is what kills the cat. Otherwise; we are always alive’: I’m not sure how this applies to your argument.

Page 12, last paragraph, ‘the capacity to always observe life and live is what our Christian religion calls salvation’: This assumes that the reader is Christian. It’s also debatable whether this is what every Christian would understand by salvation.

Page 13, last paragraph, ‘quantum physics agrees that cosmic wholeness is the vessel that contains life’: This seems a rather sweeping statement (‘quantum physics agrees’) but also one which seems uncontroversial. I’m also not clear how it helps your argument.  

Comments on the Quality of English Language

The English is, for the most part, very clear.  

Author Response

Thank you for the constructive observations and comments. Most of the issues you have raised will be addressed.

  1. I suggest in this paper that life eternal = salvation. And I use them interchangeably. While the word salvation may indicate Christianity, I believe eternal life is inherent to all religions. Therefore, I argue that we will live eternally through the experience of the cosmotheandric whole.
  2. Regarding the comment on page 13, I think I will make my argument as straightforward as possible from the beginning. My goal here is to prove the importance of wholeness and reject fragmentation in pursuing life and its fullness in eternity.   
  3. The prompt for this special edition of Religions is Love Science Discover the Divine. The science part is, therefore, necessary for the article to have a chance of being included in this edition. But it is not a forced part since I am just painting the life-giving importance of wholeness in Ubuntu, Paul, and Science.

Reviewer 3 Report

Comments and Suggestions for Authors

The author’s three-tiered approach is incentivizing. An excellent job is done on reconstructing ubuntu, Paul’s reconciliation, and the new cosmology. However, the author does not do a good job of synthesizing these views. The paper remains an excellent literature review that does not contribute to the formulation of an integrated perspective. I suggest the author pay attention to Pentecostal scholars such as Nimi Wariboko and others.

Furthermore, the interpretation of the African worldview through an Afrocentric approach appears to bestow more credit to Egypt than to the African people of the sub-Saharan content. Afrocentrism, rooted in North America, carries an implicit assumption that civilization in black Africa could only have emerged through Egypt. It is crucial for the author to recognize the distinction between Afrocentrism and the African understanding of the origin of African civilization. This perspective, however, does not align with the diverse and multifaceted nature of African civilizations across the continent, particularly neglecting the contributions of sub-Saharan cultures. Therefore, it is imperative for the author to be mindful of these nuances and refrain from attributing the concept of ubuntu to ancient Egypt, as such attribution could be perceived as perpetuating an epistemic injustice. The author should consider acknowledging the rich tapestry of African civilizations beyond Egypt and emphasize the need to approach the interpretation of the African worldviews with a broader and more inclusive perspective. This would contribute to a more accurate representation of the continent's diverse cultural and historical heritages, avoiding the potential pitfalls of oversimplification and misrepresentation inherent in a strictly Afrocentric lens.

On page 2, the statement, “Ubuntu is a South African expression adopted by most Africans…” is inaccurate. Ubuntu is a term that transcends national boundaries and is commonly shared among numerous ethnic groups in Southern and Central Africa (e.g., the Bemba people of Zambia). It is not limited to South Africa alone, and its influence extends across various cultures in the region. To provide a more precise representation, the author should consider revising the statement to accurately reflect the broad geographical and cultural scope of the concept of ubuntu. For instance, it would be more accurate to state that Ubuntu is a term embraced by many ethnic communities across Southern and Central Africa, emphasizing its widespread presence and significance beyond the borders of any specific country. This adjustment would contribute to a more nuanced and accurate portrayal of the cultural diversity associated with the concept of ubuntu ( For a detailed discussion of the history and misinterpretations of the notion of ubuntu see, Chammah J. Kaunda, The Paradox of Becoming: Pentecostalicity, Planetarity, and Africanity. London: Peter Lang, 2023).

The concepts of redemption and reconciliation are not systematically developed, representing a missed opportunity for the author to illustrate how the synthesis of the three narratives is "fundamental for life and the taste of eternal life."

In addition, there are numerous inconsistencies in footnote style and references that need to be addressed for the overall coherence of the paper.

Author Response

Your article review is quite interesting, and I will attempt to address your concerns. The bibliography provided is greatly appreciated. It will significantly serve in another project I am working on.

  1. I understand the advice to pay attention to Pentecostal scholars. However, the prompt for this issue of the Journal Religions is Love Science Discover the Divine. While African Pentecostal scholars focus on the spirit that enchants the African universe problematising the idea of becoming, I am primarily interested in elucidating the idea of wholeness in African cosmology. There is an underlying interest in becoming, but the main goal is primarily the wholeness in Ubuntu that informs African cosmology or vice versa, which informs African philosophy and anthropology.
  2. I have not set out to claim a theory of cultural monogenesis in Africa. However, in The African Origin of Civilization: Myth or Reality, Cheikh Anta Diop, on page 140, writes: “The oneness of Egyptian and Black culture could not be stated more clearly. Because of this essential Identity of genius, culture, and race, today all negroes can legitimately trace their culture to ancient Egypt and build a modern culture on that foundation.” There are a lot of African Scholars (Dr. Kgalushi Drake Koka, Nioussérê Kalala Omotunde born Jean-Philippe Corvo, etc.) who have long debunked the colonialist and hegemonic tendencies that led the West to separate Egypt from the rest of the continent. Sub-Saharan Africa does not mean a thing to Africans. It only means something to the colonial master who found in it a way of asserting power and subjugating the continent. We are one people, as the map of migration from the center of the continent to Egypt and from Egypt to the different localities of the continent from Diop demonstrates (see the map below). Cultures grow, develop, and change. There is a diversity of African cultures. However, we are one people belonging to the same continent.

3. Thank you for mentioning that Ubuntu is not just a South African concept. I should have rather stated that it is a Bantu word and/or concept adopted by the continent as representative of its worldview.

Round 2

Reviewer 3 Report

Comments and Suggestions for Authors

While I do not agree with the author's homogeneous conception of African people and their uncritical and unqualified leaning towards Egypt (Afrocentrism fundamentally revolves around Egypt-centricity. It reduces African achievement to Egypt as the symbol of African civilization), the author has addressed most of the concerns, and the manuscript is now acceptable in its current form for publication.

Comments on the Quality of English Language

The language is fine.